# Antioxidant Food Supplementation in Cancer: Lessons from Clinical Trials and Insights from Preclinical Studies

**DOI:** 10.3390/antiox14101261

**Published:** 2025-10-20

**Authors:** Alessandra Pulliero, Barbara Marengo, Oriana Ferrante, Zumama Khalid, Stefania Vernazza, Nicolò Ruzzarin, Cinzia Domenicotti, Alberto Izzotti

**Affiliations:** 1Department of Health Sciences, University of Genoa, 16132 Genoa, Italy; oriana.ferrante@edu.unige.it (O.F.); or zumama.khalid@usherbrooke.ca (Z.K.); nicolo.ruzzarin@edu.unige.it (N.R.); 2Department of Experimental Medicine, University of Genoa, 16132 Genoa, Italy; barbara.marengo@unige.it (B.M.); stefania.vernazza@unige.it (S.V.); izzotti@unige.it (A.I.); 3IRCCS Ospedale Policlinico San Martino, 16132 Genoa, Italy; 4Faculty of Medicine and Health Sciences, University of Sherbrooke, Sherbrooke, QC J1N 3C6, Canada; 5Centro 3R, Department of Information Engineering, University of Pisa, Largo Lucio Lazzarino 1, 56122 Pisa, Italy

**Keywords:** antioxidants, oxidative stress, cancer prevention, cancer relapses, food supplementation in cancer

## Abstract

Food antioxidant supplementation has been widely proposed for cancer prevention and adjuvant therapy due to the pleiotropic role of antioxidants. Herein, particular attention is given to recent clinical trials based on the use of dietary supplements in cancer patients, both as monotherapy and in combination with standard treatments, exploring both their potential benefits and risks. This review focuses on the efficacy of the most important food antioxidants, highlighting how their action may change depending on different factors such as cancer type, dose, timing of administration and antioxidant status of the patient. The results of clinical trials are often contradictory, and the clinical benefit of dietary antioxidants appears more consistent in patients with a baseline antioxidant deficiency. Furthermore, by analyzing the mechanisms underlying the contradictory clinical evidence and critically addressing the issues related to the methodologies used in preclinical models, this review could be helpful in guiding the personalized use of antioxidant supplementation in cancer patients.

## 1. Introduction

Genetic, epigenetic, and lifestyle-related factors may promote tumorigenesis and malignant transformation as a consequence of aberrant redox homeostasis, leading to the impairment of antioxidant defenses [1]. Therefore, the use of antioxidants has been proposed as a possible approach to counteracting the onset of the carcinogenesis process [2]. In fact, several in vitro and in vivo findings and epidemiological studies have demonstrated that antioxidants, by either quenching Reactive Oxygen Species (ROS) or enhancing DNA repair enzyme activity, can help prevent certain kinds of cancer [3]. However, evidence from randomized controlled trials often conflicts due to the heterogeneity of types and doses of antioxidants tested and the timing of supplementation [4,5,6,7,8]. Moreover, another critical yet often overlooked factor contributing to the inconsistent and sometimes conflicting results regarding the efficacy of antioxidant supplementation in cancer treatment is related to the experimental conditions under which in vitro studies are conducted. In fact, the composition of standard culture media differs substantially from that of human plasma, both in nutrient concentration and in the presence of metabolites. These non-physiological conditions can significantly alter cellular responses, including redox balance and metabolism, thereby potentially confounding the observed effects of antioxidant compounds. Therefore, the use of conventional media without consideration of their physiological relevance may limit the translational value of in vitro findings and contribute to the observed discrepancies in preclinical and clinical research.

Indeed, many studies have demonstrated that people receiving antioxidant supplementation have developed cancer at a higher rate or the same rate than those treated with a placebo, raising the possibility that these supplements may be harmful rather than beneficial [9,10,11,12]. One possible explanation is that antioxidants can change their biological function based on the lifestyle habits (e.g., smoking) of the subjects involved, paradoxically promoting instead of reducing pro-oxidative products, which can lead to a higher risk of developing cancer [13].

To date, many studies concerning these compounds have been published, but, to the best of our knowledge, none of these studies have examined all of them together. The primary aim of this review is to critically evaluate the role of pleiotropic antioxidant supplements in modulating cancer development and progression. Particular attention is given to recent clinical studies assessing the efficacy of these supplements, both as standalone interventions and in combination with conventional therapies. To provide a comprehensive perspective, we have also considered relevant preclinical in vitro and in vivo studies that help elucidate the underlying molecular mechanisms and cellular responses induced by antioxidant treatment. Furthermore, by integrating clinical evidence with mechanistic insights and methodological considerations, this review aims to offer a more accurate and translationally relevant understanding of antioxidant supplementation in cancer therapy.

## 2. Effects of Food Supplementation in Cancer Treatment: An Update on Clinical Trials

Identification of the real potential and limitations of antioxidant food supplementation from a clinical point of view is essential. To provide such information, this section has been focused on examining clinical trials conducted from 1999 on cancer patients treated with antioxidants and whose results have been published in high-impact journals. For completeness, two case studies were also included.

Based on the results, these studies were divided into three subgroups: (1) those in which the approach is effective in counteracting cancer progression (Table 1); (2) those in which the approach is not able to counteract cancer progression (Table 2); and (3) those in which the supplementation, despite not reducing tumor mass, can improve the patients’ quality of life (Table 3).

By analyzing the first and the second group of studies, it is evident that the response to antioxidant supplementation is closely specific to cancer type (Table 1 and Table 2). Moreover, although the studies reporting an effective action of supplementation on cancer remission are promising for some types of cancer, the small number of patients treated negatively impacts their clinical value (Table 1).

However, it is important to consider that most of the supplements leading to a positive effect on cancer remission may, under other conditions (depending not only on cancer type/cancer stage but also on the dose and the time of antioxidant administration), show null or sometimes even negative effects (Table 2).

**Table 2 antioxidants-14-01261-t002:** Clinical trials in which antioxidant supplementation was ineffective in counteracting cancer progression. The efficacy of supplemental antioxidants was evaluated and quantified by comparing clinical outcomes between patients treated with the antioxidants and those receiving placebo or standard care, as reported in the original clinical studies.

Cancer Type	Antioxidants	Conventional Therapy	Patients Enrolled (No.)	Ref.
Breast cancer	- Melatonin (20 mg starting the night before their first RT, continuing throughout RT and for an additional 2 weeks following completion of RT)	RT	79 (early-stage or ductal carcinoma in situ)	[38]
	- Vitamin D (10,000 IU and 1000 mg of calcium each day for 4 months)	Biphosphonate	40 (bone metastatic breast cancer)	[39]
	- Curcumin (6 g/day for 7 days)	Docetaxel (100 mg/m^2^) every 3 weeks for six cycles with methylprednisolone (50 mg, six times in 3 days)	42 (HER2-negative metastatic or loco-regionally recurrent or inoperable patients)	[40]
	- Vitamin D (1000 IU/day) alone or in combination with probiotic (*Lactobacillus casei*, *L. acidophilus*, *L. rhamnosus*, *L. salivarius*, *L. reuteri*, *Bifobacterium lactis*, *B. longum*, and *Bifidum*, each at 1 × 10^9^ CFU/g) + fructo-oligosaccharide (38.5 mg) for 18 weeks)	Not specified	76 (T1-3, No-2, non-metastatic)	[41]
Colorectal cancer	- Flavonoids (median intake of 196.4 mg/d) - Lignans (median intake of 0.7 mg/d)	-	409 (I–IV)	[42]
	- Vitamin D (2000 IU daily for 46 months)	-	72 (metastatic colorectal cancer)	[43]
Head and neck cancer	- Glutamine (10 g/3 times/day)	CT	40 (locally advanced cancer	[44]
	- NAC (600 mg daily for 2 years) with or without Vit A (300,000 IU daily for 1 year followed by 15,000 IU for a 2nd year)	-	2592 (non-small-cell lung cancer, stages pT1-2, N0-1, and T3N0; cancer of larynx, stages Tis, T1-3, N0-1; cancer of the oral cavity, stages T1-2 and N0-1)	[45]
	- Vitamin E (400 IU/day) + β-carotene (30 mg/day) during RT and for additional 3 years	RT	540 (stage I–II)	[46]
Myeloma	- Resveratrol (5 g for 20 days in a 21-day cycle up to 12 cycles). After two cycles, those with stable disease received two additional cycles; those with progressive disease had bortezomib	Bortezomib (1, 3 mg/m^2^ on days 1, 4, 8, and 11)	24 (multiple myeloma)	[47]
	- Vitamin C (70 mg/kg daily until discharge; after discharge, patients received oral vitamin C twice a day in two capsules of 500 mg until day 42 of the study)	-	44 (multiple myeloma or lymphoma)	[48]
Non-acute promyelocytic leukemia	- Vitamin C (1 g/day over 30 min after arsenic trioxide)	Arsenic trioxide (0.25 mg/kg/day over 1–4 h, for 5 days a week for 5 weeks)	10 (relapsed or refractory acute myeloid leukemia, excluding acute promyelocytic leukemia)	[49]
Nonmelanoma skin cancer	- Capsule containing Vit C (500 mg), Vit E (400 IU), zinc (50 mg) every day for 60 days	Surgery	60 (stage n.d.)	[50]
Non–muscle-invasive bladder cancer (NMIBC)	- Oral selenium (200 μg/d of high-selenium yeast) + vitamin E placebo - Vitamin E (200 IU/d of d-alfa-tocopherol) + selenium placebo - Selenium (200 μg/d) + vitamin E (200 IU/d) - Placebo and placebo	-	270 (Ta, Tis, T1 at baseline)	[51]
Prostatic cancer	- Vitamin C (weekly infusions: week 1, 5 g; week 2, 30 g; and weeks 3–12, 60 g) followed by efficacy evaluation at 12 weeks	-	23 (chemotherapy-naive metastatic castration-resistant cancer)	[52]
	- Lycopene (35 mg), selenium (55 mg), and green tea catechins (600 mg) for 6 months	-	60 (multifocal high-grade intraepithelial neoplasia and/or atypical small acinar proliferation)	[10]
	- Vitamin E (50 mg/day for 5–8 years until death or trial closure)	-	29,133 (stage n.d.)	[53]
Other malignancy	- Vitamin C (0.4–1.5 g/kg three times/week)	Not specified	24 (stage n.d.)	[54]
	- Vitamin C (1 g/min for 4 consecutive days/week for 4 weeks, starting at 30 g/m^2^ in the first cohort; for subsequent cohorts, dose was increased by 20 g/m^2^ until a maximum tolerated dose was found)	Imatinib (400 mg once daily)	17 (refractory to standard therapy)	[55]
	- Vitamin D (60,000 IU once weekly for an initial 8 weeks along with imatinib)	-	62 (chronic-phase chronic myeloid leukemia)	[56]
	- Vitamin E (200 IU/day with or without selenium 200 µg/day for 18 months)	-	720 (non-muscle-invasive bladder cancer)	[51]

Considering the clinical trials reported in Table 1 and Table 2, antioxidant supplementation appears to have a beneficial effect in patients with antioxidant deficiencies, while it appears to have a detrimental effect in patients with adequate antioxidant status. Therefore, the evaluation of blood antioxidant levels could be helpful in identifying patients who could be potentially treated with antioxidant supplements.

However, the results obtained from clinical studies conducted to date are extremely variable because of the heterogeneity among the populations examined, the variability in the measurement of analytes, and the poor standardization of the experimental methods used. Therefore, the interaction between the multiple cofactors makes it essential to apply more complex statistical analyses, such as multivariate analysis, to more correctly identify the possible relationships between dietary supplementation and the patient’s response to cancer. A further factor that could justify the high heterogeneity of the results is the fact that dietary supplements are considered to be as cofactors indirectly involved in carcinogenesis.

Furthermore, unbalanced antioxidant supplementation, interfering with overlapping metabolic pathways, could make it difficult to identify a single causal agent since it adds many variables that could explain the discrepancies in the results obtained.

Similar doubts arise about the beneficial effect of administering antioxidants in combination with anti-cancer therapies. In fact, although several pre-clinical studies have demonstrated that antioxidant supplementation may potentiate the effect of anti-cancer therapies, the results obtained by clinical trials are often conflicting [1]. Indeed, on one hand, the administration of supplements has been found to counteract the effects of chemo- and radiotherapy, which exert their cytotoxic action by inducing oxidative stress through the depletion of antioxidant reserves [57]. On the other hand, high doses of supplements often determine a pro-oxidant effect rather than an antioxidant one [58]. In addition, since therapy-resistant tumors are characterized by increased antioxidant levels, their supplementation could promote therapy refractoriness [57,59,60,61,62,63,64]. Furthermore, supplements can interfere with the physiological action of ROS and with the biodistribution of chemotherapeutic drugs [58].

In a limited number of clinical studies, the combination of chemo-/radiotherapy and antioxidants has been tested, and the current evidence indicates that it is better to administer antioxidants only at the end of therapy. However, it is necessary for these food supplements to selectively interfere with ROS production by cancer cells without altering their physiological role as cell signaling molecules.

Moreover, while antioxidant supplementation may not always lead to measurable improvements in tumor response or survival outcomes, a growing body of evidence suggests that these compounds can play a valuable supportive role by alleviating the debilitating side effects associated with conventional cancer therapies (Table 3). These adverse effects—often underestimated—can significantly impact patient quality of life, reduce treatment adherence, and compromise overall therapeutic success. By mitigating toxicity and improving tolerability, antioxidants may enhance patient resilience, psychological well-being, and motivation to complete intensive treatment protocols, thereby indirectly contributing to better clinical outcomes.

**Table 3 antioxidants-14-01261-t003:** Clinical trials in which antioxidant supplements have a positive (+) or negative (-) impact on therapy side effects and patients’ quality of life. The efficacy of supplemental antioxidants was evaluated and quantified by comparing the results obtained in antioxidant-treated patients with those obtained in placebo groups.

Cancer Type	Antioxidants	Conventional Therapy	Patients Enrolled (No.)	Effect	Ref.
Acute lymphoblastic leukemia	- Curcumin	Vincristine	141 pediatric patients	+	[65]
	- Curcumin (180.34 ± 9.46 mg per capsule) - Resveratrol (64.51 ± 6.08 mg per capsule) - Lignans (32.44 ± 0.81 mg per capsule) - Isoflavones (17.80 ± 0.99 mg per capsule) - Hydroxycinnamic acid derivatives (1.34 ± 0.18 mg per capsule)	None; candidate for mastectomy	39 (Tis-IIb)	+	[15]
	- Flavonoids (30.62 ± 20.99 mg/day in subjects with cancer recurrence) (57.11 ± 41.08 mg/day in subjects without cancer recurrence)	RT and CT	572 (0-IIb)	+	[17]
Breast cancer	- Curcumin	RT	52	+	[66]
	- Vitamin C (Pascorbin^®^ 7.5 g for at least 4 weeks)	CT and RT	53 (stages Iia–IIIb)	+	[67]
	- Vitamin C (500 mg) and vitamin E (400 mg) for 5 months	Fluorouracil, doxorubicin, cyclophosphamide	40 (stage II)	+	[68]
	- Melatonin (20 mg starting the night before their first RT, continuing throughout RT and for an additional 2 weeks following completion of RT)	RT	79 (early stage or ductal carcinoma in situ)	-	[38]
	- Melatonin (3 mg for 4 months)	-	95 (stages 0–III)	+	[69]
	- Melatonin (1 mg/day for 3 months)	CT	49 (stage n.d.)	+	[70]
	- Vitamin D (10,000 IU) + calcium (1000 mg) daily for 4 months	-	40 (bone metastatic breast cancer)	-	[39]
	- Vitamin D (0.5 µg once daily during the whole treatment)	Doxorubicin (60 mg/m^2^) + Cyclophosphamide (600 mg/m^2^) every 21 days	150 (stage n.d.)	+	[71]
	- Vitamin D (2000 IU/kg) + calcium (4000 IU/kg) for 12 weeks	Letrozole	82 (stage n.d.)	+	[72]
Colorectal cancer	- Curcumin (2 g/d)	FOLFOX	27 (metastatic colon cancer)	+	[22]
	- Flavonoids (n.d.)	-	2552 (stage I–III)	+	[73]
	- Vitamin D (50,000 IU soft gel daily) with or without omega-3 fatty acid capsules daily for 8 weeks	CT	81 (stage II–III)	+	[74]
Hepatocellular carcinoma	- Coenzyme Q10 (300 mg/day for 12 weeks)	Surgery	41 (primary hepatocellular carcinoma)	+	[75]
Leukemia	- Vitamin C (50–80 mg/kg/day, days 0–9)	DCAG (15 mg/m^2^ of decitabine (days 1–5) and 300 μg/day of granulocyte colony-stimulating factor (G-CSF, days 0–9) for priming its combination with 10 mg/m^2^ of cytarabine (q12h, days 3–9), 8 mg/m^2^ of aclarubicin (days 3–6)	73 (acute myeloid leukemia)	+	[33]
	- Vitamin A (180,000 IU)	Methotrexate (3 g/m^2^ (leukemia) and 5 g/m^2^ (Lymphoma)	35 (leukemia and lymphoma)	+	[76]
LHNC (locally advanced head and neck cancer)	- Curcumin (1 g every 8 h)	EBRT	60 (stage III–IVB)	+	[30]
Melanoma	- Vitamin D (100,000 IU monthly)	Surgery	500 (stage IB–III)	+	[24]
Multiple myeloma	- Zinc (30 mg daily for 1 month)	CT	36 (patients in complete response status and candidates for autologous hematopoietic stem cell transplantation)	+	[77]
Nonmelanoma Skin cancer	- Capsule containing vit. C (500 mg) + vit. E (400 IU) + zinc (50 mg) daily for 60 days	Surgery	60 (stage n.d.)	-	[50]
Ovarian cancer	- Indole-3-carbinol (400 mg/daily) - I3C + EGCG (400 mg/daily + 400 mg/daily, respectively)	neoadjuvant platinum-taxane, surgery, adjuvant platinum-taxane	284 (stage III–IV)	+	[34]
Thoracic cancer (breast, lung, esophageal)	- EGCG (from 660 to 2574 μmol/L sprayed on the irradiated area)	RT	19 (severe radio-induced dermatitis in cancers stage II–IV)	+	[37]
Other malignancies	- Vitamin C (intravenous administration of 10 g twice with a 3-day interval and oral intake of 1 g daily for a week)	-	39 (terminal patients)	+	[78]
	- Vitamin E (300 mg twice/day during chemo-therapy and 3 months after its cessation)	Paclitaxel	32 (stage n.d).	+	[79]
	- Vitamin E (600 mg/day during chemotherapy and 3 months after its cessation)	Cisplatin	30 (stage n.d)	+	[80]
	- Vitamin E (400 mg/day, prolonged for 3 months after chemotherapy ended)	Cisplatin (300 mg/m^2^)	23 (stage n.d.)	+	[81]
	- Vitamin E (400 mg/daily, prolonged for 3 months after suspension of chemotherapy)	Cisplatin	108 (stage n.d.)	+	[82]
	- NAC (1200 mg starting 2 days before surgery)	Surgery	33 (stage n.d.)	+	[83]

### 2.1. Curcumin

Several studies have been conducted to investigate the effect of curcumin on different solid tumors: breast, prostate, colorectal, head and neck cancer, and hepatocellular carcinoma.

The mechanisms through which curcumin seems to be a good support in anti-cancer treatment vary from its direct effects on the tumor to its coadjutant role in radio- or chemo-therapy as an antioxidant and anti-inflammation agent. Furthermore, curcumin administration seems to reduce the side effects of therapies, also improving patients’ compliance.

A prospective randomized study conducted on LHNC (locally advanced head and neck cancer) patients showed that curcumin is able to decrease chemo-radiation therapy-induced hematological, renal, skin, and mucosal toxicities resulting in improvement of disease control (Table 1) [30].

Ávila-Gálvez et al. demonstrated that curcuminoids directly affect breast cancer growth by inducing cell-cycle arrest, senescence, and apoptosis via a p53/p21-dependent pathway, while isoflavone-derived metabolites exert estrogenic-like activity, mainly in p53-wild-type MCF-7 cells, suggesting that the consumption of 3 capsules/day (turmeric, red clover, and flaxseed extracts plus resveratrol; 296.4 mg phenolics/capsule) could be helpful in fighting breast cancer [15].

Curcumin is also a safe and tolerable adjunct to chemotherapy (FOLFOX ± bevacizumab vs. FOLFOX ± bevacizumab plus curcumin, CUFOX) in patients with metastatic colorectal cancer. Despite the low number of patients included in the study, CUFOX patients had smaller negative changes to their functional, symptom, and global health scores. In addition, curcumin has been found to prevent oxaliplatin-induced neuropathy in CUFOX patients (Table 1) [22].

Moreover, a study carried out on HCC patients treated with trans-arterial chemoembolization (TACE) demonstrated that the combination of curcumin, piperine, and taurine (CPT) significantly increased IFN-γ serum levels (Table 1) [28], downregulated the expression of PD-1, and induced a significant reduction in Treg lymphocytes. Moreover, the CPT approach is clinicopathologically relevant, since it has been found to exert a high impact on aspartate aminotransferase (AST), lactate dehydrogenase (LDH), and alpha fetoprotein (AFP) levels, which significantly declined after treatment.

A randomized, double-blind, placebo-controlled, phase II presurgical trial conducted by Macis et al. on patients with adenomatous polyps demonstrated that the combination of anthocyanins and curcumin did not directly modulate the circulating biomarkers of inflammation and metabolism, but it influenced the metabolic biomarkers involved in colon carcinogenesis via adiponectin and IL-6 levels, since these molecules influence the tumor microenvironment and its energetic balance (Table 1) [14].

### 2.2. Trans-Resveratrol

Trans-resveratrol is present in high concentrations in the skin of red grapes and red wine and has been advertised as a chemopreventive agent, since it prevents the formation of mammary tumors and has a dose-dependent antiproliferative effect in vitro. A study by Zhu et al. [84] showed the effects of trans-resveratrol on women with increased breast cancer risk vs. a placebo-treated group. Significant increases were found in serum total resveratrol and its glucuronide metabolites (*p* < 0.001), while a reduction in breast NAF PGE_2_ levels correlated with RASSF-1α demethylation (*p* = 0.045). Trans-resveratrol supplementation can affect epigenetic regulation and inflammatory markers in breast tissue, suggesting a promising chemopreventive mechanism. Further large-scale studies are recommended.

On the other hand, in a phase II clinical trial conducted in patients with relapsed/refractory multiple myeloma, it was observed that the combined therapy of resveratrol plus bortezomib had an unacceptable safety profile and minimal efficacy [47].

Recently, in an open-label study conducted in patients with head and neck cancer and receiving home enteral nutrition, it was reported that resveratrol supplementation enhanced antioxidant defenses by increasing glutathione peroxidase activity and superoxide dismutase protein levels [27].

### 2.3. Selenium and Vitamin E

Selenium supplementation (SS) was investigated for the prevention of prostate cancer, while selenium and vitamin E supplementation was investigated for the prevention of non-muscle-invasive bladder cancer (NMIBC) relapse.

A subsequent report analyzed the correlation between prostate-specific antigen (PSA) and selenium concentrations, showing that SS significantly reduced the incidence of prostate cancer and that this effect was greater in men with the lowest baseline plasma selenium concentrations (Table 1) [35].

A multi-center, prospective, double-blinded, placebo-controlled clinical trial including patients with newly diagnosed NMIBC was carried out by Bryan et al. [51]. The results showed that SS did not reduce the risk of recurrence in NMIBC patients, while vitamin E supplementation was associated with an increased risk of recurrence. No effect due to selenium or vitamin E supplementation was observed on cancer progression or overall survival, even if vitamin E supplementation appeared to be harmful for NMIBC patients.

The results obtained with vitamin E supplementation are extremely heterogeneous (Table 1, Table 2 and Table 3). In fact, for example, in a randomized controlled trial on patients affected by head and neck cancer and treated with α-tocopherol and β-carotene during radiation therapy, vitamin E was demonstrated to be harmful [46]. On the other hand, it was reported that vitamin E administration, given alone or in combination with β-carotene, increased the overall survival of prostate cancer patients [53].

### 2.4. Vitamin A

In a study conducted in patients with advanced cervical cancer, vitamin A supplementation was found to increase the clinical response to cisplatin and paclitaxel (Table 1) [20]. Furthermore, it was interesting to note that vitamin A administration, before methotrexate treatment, appeared to be effective in protecting children affected by leukemia and lymphoma from drug-induced D-xylose malabsorption (Table 3) [76].

A robust randomized clinical trial was performed in 18,314 subjects at risk for lung cancer, treating them with vitamin A for 4 years (Beta Carotene and Retinol Efficacy Trial). The results provided evidence that no benefit was obtained in terms of cancer prevention. Furthermore, vitamin A supplementation was associated with increased mortality from lung cancer and cardiovascular disease. In addition, beta carotene appeared to alter the redox balance, acting as a pro-oxidant agent in non-physiologic conditions [85].

### 2.5. Vitamin D

Contradictory results have also been observed in clinical trials in which the efficacy of vitamin D was tested. In fact, as reported in Table 1, vitamin D administration to patients with cervical intraepithelial neoplasia was found to have beneficial effects on CIN1/2/3 recurrence and metabolic status [21]. Similar positive effects have also been detected in patients with gastric cancer [25] and melanoma [24]. Positive results have also been observed in clinical studies in which vitamin D was combined with conventional therapy. In fact, in a phase II study conducted on breast cancer patients, vitamin D at high doses was found to enhance the action of conventional chemotherapy [18]. Similarly, vitamin D supplementation was observed to increase the progression-free survival rate in colorectal cancer patients treated with mFOLFOX6 plus bevacizumab [23].

In contrast, other studies have reported the ineffectiveness of vitamin D supplementation (Table 2). In fact, treatment with vitamin D3 and calcium in patients with bone-metastatic breast cancer did not produce significant beneficial effects [39]. A similar situation was described in patients with colorectal cancer treated with cholecalciferol [43] and in patients with chronic myeloid leukemia treated with Vitamin D in combination with imatinib [56]. Moreover, in a recently published study, it was found that the treatment of vitamin D increased IL-6 serum levels and, when it was administered together with symbiotic supplements, markedly enhanced the anti-inflammatory index, suggesting a potential anti-inflammatory function of this combination [41].

Furthermore, several studies have shown that vitamin D supplementation is able to improve the quality of life of different oncologic patients and to attenuate the side effects of conventional therapies (Table 3).

### 2.6. Vitamin C

The effects of vitamin C supplementation on cancer are also contradictory and complex. As reported by Zhao, the addition of vitamin C to low-dose decitabine before aclarubicin and cytarabine (DCAG) increased the rate of complete remissions and was associated with prolonged overall survival in patients with acute myeloid leukemia (Table 1) [33]. In addition, vitamin C supplementation was also shown to potentiate the biological effects of DNA methyltransferase inhibitors in patients with myeloid cancer (Table 1) [32].

In contrast, several studies have reported the ineffectiveness of vitamin C supplementation. In this regard, a study conducted on patients with prostate cancer demonstrated that intravenous administration of vitamin C did not induce disease remission and, therefore, its supplementation was not recommended (Table 2) [52]. Similar results were obtained in patients with nonmelanoma skin cancer treated with a cocktail containing vitamin C, vitamin E, and zinc, where no significant reduction in oxidative stress biomarkers was observed (Table 2) [50]. Limited clinical efficacy was also reported in patients with acute myeloid leukemia treated with a combined therapy of vitamin C and arsenic trioxide (Table 2) [49]. Other studies have demonstrated that, although vitamin C was well tolerated, it failed to demonstrate anticancer efficacy (Table 2) [54,55].

Better results have been obtained in terms of the improvement in quality of life and reduction of side effects of conventional therapy (Table 3). In several studies conducted in patients with breast cancer [68], nonmelanoma skin cancer [50], leukemia [33], and other neoplasia [78], it has been reported that vitamin C supplementation was well tolerated and led to a significant reduction in disease-induced and chemo/radiotherapy-promoted disorders.

These contradictory results are related to the fact that vitamin C behaves either as an antioxidant or pro-oxidant depending on the dose and the administration route.

Vitamin C acts as an antioxidant when administered at low doses by the oral route, where it is metabolized as ascorbic acid. This is currently the most used manner of administration and appear to be ineffective, if not detrimental, in cancer patients.

Conversely, vitamin C acts as pro-oxidant when used at high doses through blood infusion, where it is metabolized as the oxidant ossalic acid [86]. Indeed, when high doses of vitamin C are administered intravenously, most of it is metabolized to oxalic acid. Under these conditions, clinical effects in cancer patients are promising and warrant further clinical studies.

### 2.7. Flavonoids

The effects of epigallocatechin-3-gallate (EGCG) solution can be useful to treat the acute severe dermatitis that may occur after radiotherapy. A phase I study on radiotherapy-treated patients with thoracic cancer, grade III RID (Regional or Distant Involvement), showed that three days of continuous EGCG administration (sprayed in the radiation field) induced the regression of lesions to grade I or grade II RID and significantly reduced associated symptoms after 15 days of EGCG treatment (Table 1 and Table 3) [37].

Among flavonoids, phytoestrogens have been suggested to have an anti-proliferative role in prostate cancer. Ahlin et al. [87] carried out a randomized controlled trial involving men with low- to intermediate-risk prostate cancer scheduled for radical prostatectomy. Participants were randomized either to an intervention group who received ~200 mg/day of phytoestrogens (derived from soybeans and flaxseeds), added to their usual diet for approximately 6 weeks, or a control group following standard dietary advice without supplementation. Dietary supplementation with soybeans and flaxseeds (~200 mg phytoestrogens/day) did not affect circulating levels of testosterone, IGF-1, or SHBG in the perioperative setting, even though a high intake of phytoestrogens was hypothesized to reduce the concentration of estradiol in prostate cancer patients with a specific genetic profile of Erβ.

Anticancer effects of flavonoids are known, but little information about the influence of flavonoid intake on colorectal cancer (CRC)-related mortality is available. A prospective study investigated the association of post-diagnostic flavonoid intake with CRC-specific and all-cause mortality in 2552 patients diagnosed with stage I-III CRC. Total flavonoid intake was not associated with mortality, except for flavan-3-ols, a component of tea, where higher intake showed a linear relationship with lower CRC-specific and all-cause mortality (Table 3) [73].

Another study investigated the effects of flavonoid and lignan intakes on the risk of CRC recurrence and overall survival in CRC patients. The results did not demonstrate a role of flavonoid or lignan intake on CRC prognosis nor mortality, since after 8.6 years of follow-up, 32.5% participants died, and 24.1% had CRC recurrence (Table 2) [42].

While the association between flavonoids and the incidence of breast cancer is clear, less is known about the association between the intake of flavonoids and cancer recurrence. A study by Cheon et al. showed that the intake of flavonoid-rich food improved disease-free survival among overweight and obese patients, suggesting that the intake of flavonoids could have beneficial effects on cancer recurrence in these patients (Table 1 and Table 3) [17].

### 2.8. Indole-3-Carbinol (I3C)

The results obtained by a five-year study on patients with stage III-IV serous ovarian cancer demonstrated that the combination of I3C and EGCG was able to prolong progression-free survival and overall survival of patients, suggesting that long-term usage of this approach represents a promising maintenance therapy in the treatment of patients with advanced ovarian cancer (Table 1 and Table 3) [34].

### 2.9. Butyric Acid and Short-Chain Fatty Acids

There are 12 registered clinical trials (clinicaltrials.gov) currently underway, investigating the administration of butyric acid and short-chain fatty acids to cancer patients for therapeutic purposes or to cancer survivors for preventive purposes. Butyrate is generated by gut microbial fermentation of dietary fibers. Its antineoplastic effects are exerted via the activation of G protein-coupled receptors and the inhibition of histone deacetylases [88]. Despite the strong experimental and mechanistic evidence, robust results of large-sized clinical trials are not yet available.

### 2.10. Ozonated Fatty Acids

The oxidation of cancer cells has been proposed as a possible radical therapy to treat cancer cells and prevent cancer relapses [89]. Indeed, the chemo- and radioresistance of cancer cells, particularly of stem cancer cells, is due to the high amounts of their antioxidants [90]. Oxidizing agents, such as ozone, have been demonstrated to selectively inhibit the growth of human cancer cells [91]. However, a clinical study using ozone blood perfusion or insufflation obtained only limited clinical benefit, mainly focused on the attenuation of fatigue syndrome in cancer patients [92]. This limited clinical result has been attributed to the fact that when administered as gas, the persistence and penetration of ozone inside the body is limited. Indeed, the amount of gas dissolved in a body fluid is inversely related to the temperature (Henry’s law), which is relatively high in the human body (37 °C). To address this problem, the binding of ozone to a lipid carrier was proposed [93]. This approach resulted in the production of ozonated fatty acids or ozonated oils. However, anticancer properties are exerted by ozonated oils only when a sufficient amount of ozone (i.e., >800 mEq/l) is bound to the lipid carrier [94]. Furthermore, the direct oral administration of ozonated oil is not feasible as it results in poor absorption and side effects such as nausea and diarrhea. These side effects may be avoided by properly adsorbing the ozonated oil onto an adsorbing powder under slow release. High-ozonide oils were demonstrated to be effective in killing cancer cells by inducing apoptosis through oxidative damage to the mitochondrial membrane [94]. Clinical application in cancer patients has been very promising, resulting in a dramatic decrease in the risk of cancer relapses and improving clinical outcomes in a clinical study performed in 115 patients affected by solid tumors [94]. Furthermore, oxygenation of healthy tissue improves the quality of life of cancer patients and attenuates side effects of chemotherapy, such as fatigue syndrome and bone marrow suppression [94].

## 3. Mechanisms Involved in the Effects of Food Supplementation in Cancer Prevention and Therapy

Although the consumption of dietary antioxidants is increasingly encouraged, only a limited number of them can be considered for their therapeutic efficacy, and the collected results are often contradictory.

Furthermore, as emphasized by Saeidnia and Abdollahi [95], natural antioxidants may provide potential benefits in cancer chemoprevention, primarily by counteracting oxidative stress and inflammation. However, their role in cancer therapy, especially during the early stages of cancer development, remains controversial due to inconsistent and often inconclusive evidence. Therefore, rigorous and well-designed clinical trials are essential to determine the safety and therapeutic relevance of various phytochemicals in both cancer prevention and treatment [96].

Several factors may account for the disappointing outcomes of many clinical studies involving antioxidant supplementation, such as (i) the use of non-specific and poorly characterized compounds, which limits the ability to target defined molecular pathways; (ii) the administration of pharmacological doses rather than physiologically relevant dietary levels, which may alter expected biological effects; (iii) short durations of administration, which are insufficient to observe long-term preventive or therapeutic effects; (iv) poor patient compliance to supplementation protocols, which reduces statistical power; and (v) inadequate limited follow-up periods, which hinder the assessment of sustained clinical outcomes or late-onset effects. Indeed, this lack of success should encourage us to look for new ways to prevent and/or treat cancer by modulating oxidative balance. In fact, this “push” is justified by the promising results obtained in several in vitro and in vivo studies, which demonstrate a strong link between oxidative stress and cancer.

However, it is necessary to underline that the action of food antioxidants is extremely complex, and the effects are not predictable considering only the oxidative mechanisms. As summarized in Figure 1, this treatment can result in metabolic and phenotypic changes due to their effects on (i) microbiota; (ii) the immune system; (iii) ROS-sensitive signaling pathways; (iv) cell metabolism; and (v) the epigenome.

### 3.1. Effects on Microbiota

The intricate relationship between dietary antioxidants and gut microbiota unveils a novel frontier in cancer treatment, offering both opportunities and challenges for precision nutrition interventions [97]. Among these antioxidants, resveratrol and curcumin stand out for their impact on gut microbial composition and functionality, thereby influencing cancer progression and relapse risk [98,99].

Resveratrol exerts potent anticarcinogenic effects by reshaping the gut microbiota landscape. Clinical trials performed by Singh and Zhu et al. reported the therapeutic outcomes of resveratrol in breast, colorectal cancer, and multiple myeloma. Resveratrol was found to suppress methylation of the RASSF-1α gene and to lower breast cancer-promoting PGE2 levels. In contrast, in colon cancer patients, micronized resveratrol (SRT501) was found to increase the expression of cleaved caspase-3 in malignant hepatic tissue, indicating potential apoptosis in colorectal cancer patients. The 39% increase in cleaved caspase-3 strongly indicates enhanced apoptosis in malignant hepatic cells [84,100]. Resveratrol has been found to create a gut environment conducive to cancer prevention and therapy by promoting the growth of beneficial microbial species while inhibiting the proliferation of pathogenic bacteria. Moreover, its anti-inflammatory properties mitigate tumor-promoting inflammation, further attenuating cancer relapse risk [101].

EGCG has excellent antioxidant properties, and it also maintains gut microbiota stability and health by inhibiting pathogenic bacteria proliferation and by increasing the number of probiotic bacteria [102].

Genistein belongs to the isoflavone class and can modulate the abundance of phenolic metabolites produced by gut microbes, modifying the metabolism of gut microbiota [103]. Another source of phenolic compounds is *Hibiscus sabdariffa* L., which was recently found to modulate the intestinal microbiota (by increasing the amount of *Lachnospiraceae*, *Ruminococcaceae*, *Clostridiaceae*, and the genus *Clostridum*, important producers of butyrate) and to induce apoptosis during the initial phase of an in vivo model of colorectal cancer [104].

Brazil nuts are a rich source of selenium, but their effect on gut health is still not clear. A systematic review with meta-analysis reported randomized controlled trials on nut consumption, and significant increases in the gut contents of *Dialister*, *Clostridium*, *Roseburia*, and *Lachnospira*, as well as a significant decrease in Parabacteroides, were described [105].

Similarly, curcumin, derived from turmeric, modulates the gut microbiota composition, offering promising avenues for cancer management. Through its antimicrobial activity, curcumin suppresses the growth of harmful bacteria implicated in carcinogenesis while fostering the proliferation of beneficial strains, such as *Ruminococcin* [106,107]. This rebalancing of the gut microbiota ecosystem aligns with reduced inflammation and enhanced immune surveillance against cancer cells, underscoring curcumin’s potential as an adjuvant therapy in cancer treatment [108,109]. A similar study performed by Liu et al., 2022, also observed a sensitization response of curcumin in acute myeloid leukemia through the regulation of intestinal microbiota [110].

Furthermore, the symbiotic relationship between dietary antioxidants and the gut microbiota extends beyond mere compositional changes. These antioxidants, by promoting microbial metabolism of bioactive compounds, generate metabolites with potent anticancer properties in colorectal cancer [111]. Flavonoids, a subclass of dietary antioxidants, undergo microbial metabolism in the gut, yielding metabolites with demonstrated anticarcinogenic effects. Through the modulation of gut microbial activity, flavonoids contribute to the maintenance of gut homeostasis, thereby mitigating cancer progression and relapse [84]. In addition, it has been recently demonstrated that the combined treatment of sulforaphane (an antioxidant extracted from broccoli sprouts) and inulin successfully inhibited in vivo breast cancer growth and increased tumor onset latency through several mechanisms, including changes in gut microbiota composition, by increasing *Ruminococcus*, *Muribaculaceae*, and *Faecalibaculum* and decreasing the abundance of *Blautia*, *Turicibacter*, and *Clostridium sensu stricto 1* [112].

However, while the potential of dietary antioxidants in cancer treatment is promising, challenges persist in understanding the precise mechanisms underlying their interactions with the gut microbiota. Unravelling the complex interplay between dietary antioxidants, gut microbial dynamics, and cancer treatment requires comprehensive research efforts. Addressing critical questions regarding the optimal dosage, timing, and specific microbial targets of dietary antioxidants will be paramount for advancing precision nutrition strategies in cancer management [97,113].

### 3.2. Effects on Immune System

Unfortunately, studies reporting the relationships between immune system activity and antioxidant supplementation are limited. The scarcity of such data derives from multiple factors including the high time and financial costs of conducting analyses and the lack of standardized tests capable of evaluating these supplements in a specific and reliable manner.

Since micronutrient deficiencies result in deleterious outcomes in chronic diseases such as cancer, antioxidant supplementation may play a crucial role in the response to metabolic and immunological stress conditions of oncologic patients. However, to clarify the potential beneficial effects of such supplementation, it is urgently necessary to understand the specific mechanisms of action.

Several studies have investigated the role of antioxidant supplementation in cancer prevention, although the findings are often contradictory. Therefore, it is crucial to consider the diverse array of antioxidants and their multifaceted effects on the immune system and cancer relapse risk in order to determine a correct approach to cancer management.

#### 3.2.1. In Vitro Studies

Artemisinin, a derivative of Artemisia annua Linnè, is considered an antioxidant with antitumor properties [114,115]. Recently, it has been shown that artemisinin exerts its antitumor action by upregulating, on the surface of endometrial carcinoma cells, the expression of CD155, a potent immune ligand capable of stimulating the cytotoxic activity of NK cells [116].

Studies by Yan and collaborators demonstrated that hepatocarcinoma cells treated with resveratrol showed reduced expression of IL-6 and CXCR4 receptor due to the inhibitory action of resveratrol on the expression of GLI-1 (glioma-associated oncogene 1) [117]. In this context, it was shown that both CXCR4, a receptor implicated in the stimulation of chemotaxis, and IL-6, associated with the inflammatory response, are downstream effectors of GLI-1 [118]. These in vitro studies suggest that GLI-1 may be a therapeutic target useful for counteracting angiogenesis and the metastatic potential of cancer cells.

Interestingly, in a study carried out through label-free quantitative proteomic methods and real-time cellular analyses, co-treatment of breast cancer cells with vitamin C and doxorubicin was able to reduce the expression of HSP90AA1. This heat shock protein, being able to influence both the innate and adaptive immune responses, could potentially impact the clinical response to immunotherapy [119].

Finally, it has also been shown that vitamin E can induce changes in immune responses. For example, vitamin E has been found to (i) activate naïve CD8+ T cells against the antigens exposed on breast cancer and lung carcinoma cells [120]; (ii) stimulate the activation of T cells and decrease the expression of PD-L1 (programmed cell death ligand 1), a protein frequently overexpressed in cancer cells and responsible for the evasion of immune surveillance [121,122]; (iii) downregulate the expression of LCK (lymphocyte-specific kinase) involved in the activation of T-cell or B-cell receptors [123].

#### 3.2.2. Pre-Clinical In Vivo Studies

Even more interesting and promising results were observed in in vivo studies carried out to reproduce several kinds of human cancers.

In this context, it was observed that the administration of melatonin to mice with lung cancer was able to reduce metastasis formation and restore thymic efficiency [124].

Interestingly, in mice bearing melanoma lung metastasis, treatment with resveratrol was found to increase the expressions of IFN-γ and CXCL-10 [125] by activating cytotoxic T lymphocytes and NK cells, amplifying the expression of Class I MHC, inducing chemotaxis, and controlling the inflammation response. Altering the balance between regulatory T cells (Tregs) and effector T cells (Teffs) critically influences cancer progression. A higher Treg-to-Teff ratio suppresses antitumor immunity and promotes tumor growth, whereas restoring this balance in favor of effector T cells enhances immune-mediated tumor control. Since the immune system’s ability to modulate tumor growth is closely related to this immunological equilibrium, the evidence that resveratrol significantly increases activated T cells while decreasing Treg cells further supports its potential role as a useful supplement in the anticancer strategies.

Intriguingly, several studies have attributed a crucial role to vitamins in modulating innate and adaptive immune response. For example, it was found that vitamin A supplementation reduced the growth of colorectal cancer by upregulating CD8+ T cells [126]. Similar effects were observed following the administration of vitamin C to mice bearing breast, colorectal, pancreatic cancer, melanoma, and lymphoma, where reductions in tumor growth and potentiation of immune checkpoint therapy (ICT) based on anti-PD-1/PD-L1 or anti-CTLA-a antibodies were observed. Moreover, it has been shown that these responses are due to the over-production of IFN-γ and increased tumor infiltration by CD8+ and CD4+ lymphocytes [127,128]. Similarly, vitamin C counteracted ovarian cancer growth by reducing the amount of M2 macrophages in the tumoral microenvironment [129]. Likewise, vitamin D administration has been shown to reduce breast cancer growth by favoring CD8+ T cell tumor infiltration and reducing the levels of pro-inflammatory cytokines [130,131].

### 3.3. Effects on Redox-Sensitive Signaling Pathways Involved in Cell Death and Proliferation

A high number of studies have suggested that many antitumor effects of antioxidant supplementation are related to the modulation of several redox-modulated signal transduction pathways involved in cell cycle progression, cell proliferation and differentiation, apoptosis, autophagy, invasion, angiogenesis, and gap junction intercellular communication. Since the activity of these signaling pathways is altered during cancer progression, understanding the mechanisms underlying antioxidant supplementation has a crucial clinical role in the fight against cancer.

However, since a comprehensive discussion concerning the signaling pathways modulated by antioxidant supplementation is beyond the specific scope of this review, the most important studies on this topic are summarized in Table 4 (in vitro studies) and Table 5 (in vivo studies).

### 3.4. Effects on Metabolism

Several studies support the common hypothesis that the different metabolism displayed by cancer cells compared to healthy cells can be considered an “Achilles’ heel” that must be considered to develop new therapeutic strategies. In fact, one of the hallmarks of carcinogenesis is that cancer cells have basal increased levels of ROS with respect to their healthy counterparts [63,211,212,213,214,215,216,217]. Consequently, it is conceivable that antioxidant supplementation, by targeting metabolic atypia, could be useful as a therapy or as an adjuvant in anticancer treatment.

#### 3.4.1. In Vitro Studies

Melatonin is a compound useful for treating cancer, as it is able to modulate different signaling pathways involved in tumor metabolism. In fact, it has been observed that melatonin induces the inhibition of glycolysis and the stimulation of OXPHOS in prostate cancer [218] and in head and neck carcinoma cells [219].

Similar results have been observed in papillary thyroid [220], breast [162], pancreatic [163], and colorectal [166,221] cancer cells exposed to vitamin C.

A marked decrease in glycolytic enzymes and GLUT-1 receptor, with a consequent decrease in glucose uptake, was found in colorectal cancer cells [173] exposed to vitamin D and in breast cancer cells exposed to vitamin D [171] or vitamin E [183].

#### 3.4.2. In Vivo Studies

The effects that melatonin and vitamins have on metabolism have also been reported in several in vivo studies. In fact, it has been demonstrated that the administration of melatonin to mice bearing colon cancer potentiates the cytotoxic action of radiotherapy. This event is accompanied by a reduction in SOD (superoxide dismutase) and glutathione peroxidase activities and in MDA (malondialdehyde) levels [222].

Regarding vitamins, it has been shown that the treatment of mice with colon cancer using vitamin D or vitamin C induced a marked reduction of GLUT1 and PKM2 activity and expression, leading to the suppression of glycolysis [166,173].

Recently, it was found that exposure to aspirin and vitamin D, by counteracting DNA damage, was able to reduce the formation of aberrant crypts foci (ACF) in rat colon, while this did not occur following vitamin C exposure. Moreover, aspirin stimulated catalase activity, but no further activation was induced by vitamin co-treatments. On the other hand, aspirin and both vitamins markedly decreased hepatic SOD levels, but this was not accompanied by alterations in glutathione (GSH) levels [223]. Interestingly, in regard to GSH, it has been demonstrated that GSH and selenium supplementation markedly reduced the diethylnitrosamine-induced DNA damage by reducing malondialdehyde levels and by increasing GSH levels and the activities of GSH-related enzymes, such as glutathione peroxidase and glutathione- S-transferase [224].

### 3.5. Epigenetic Effects

Epigenetic changes do not induce alterations in DNA sequences, are potentially reversible, and consist of three mechanisms: DNA methylation, histone modifications, and miRNAs. miRNAs are short double-stranded RNAs that can regulate gene expression by binding to complementary mRNAs [225]. Depending on their role in cancer, miRNAs can target unsuppressed genes (oncogenic miRNAs) and oncogenes (tumor-suppressive (TS)-miRNAs).

Increasing evidence from in vitro and in vivo studies indicates that antioxidant food supplementation can influence miRNA expression. For example, melatonin was found to effectively counteract the viability and growth of thyroid cancer cells both in vitro and in vivo by reducing the expression of miRNA-21 and miRNA-30e [196]. Further studies have demonstrated that melatonin also influences the in vitro and in vivo growth of (i) hepatocellular carcinoma by inducing the expression of mi-RNA Let7i-3p [139]; (ii) triple-negative breast cancer by upregulating miRNA-152-3p [140], (iii) oral squamous cell carcinoma by over-expressing miRNA-892 a and miRNA-34b-5p [197], and (iv) glioblastoma by increasing the expression of miRNA-6858-5p [138].

Regarding the role played by vitamins in modulating miRNA expression, it was demonstrated that vitamin C inhibited colon cancer growth both in vitro and in vivo, by increasing miRNA-627 expression [226].

Furthermore, several in vitro studies have shown that resveratrol [227], vitamin A [157], vitamin D [228,229], and vitamin E [190,191] are also able of reducing the survival and inhibiting the invasion and migration of several cancer cell lines.

Among antioxidants, curcumin emerges as a standout contender due to its remarkable ability to regulate microRNAs implicated in cancer metastases [230].

Moreover, resveratrol shows a capacity to modulate histone acetylation and DNA methylation, which can potentially reprogram gene expression profiles associated with cancer relapse [99,231], presenting an attractive strategy for adjuvant cancer therapy [232].

## 4. Future Perspectives

The studies presented and discussed here highlight the clinical need to move beyond the current approach to antioxidant administration, recommending a tailored approach for each patient. This need arises from analyses of clinical trials conducted over the past 25 years, which demonstrate that antioxidant supplementation does not produce the same effects, positive or negative, in all patients. Therefore, this awareness highlights the importance of categorizing patients into those who may benefit from and those who may experience negative consequences from such supplementation. In this context, it is essential to identify valid biomarkers able to predict the patient’s response to these supplements. This goal will be achievable only through multi-center clinical trials in which nutritional and genomic data will be integrated with the patients’ lifestyles and compliance to their therapeutic plan.

Furthermore, from this perspective, it is necessary to shed light on molecular mechanisms and to focus the attention mainly on redox, immune, and metabolic mechanisms through which exogenous antioxidants can interact with therapy and the tumor microenvironment. This objective can be reached by means of proteomic, genomic, and metabolomic studies and also by 3D in vitro assays capable of mimicking the pathophysiological conditions of cancer patients.

## 5. Conclusions

The roles of nutrition and food antioxidant supplementation in cancer management is increasingly recognized, although not yet fully understood. The set of results obtained from the clinical trials reported here provides evidence that antioxidant supplementation is not always an effective strategy to counteract cancer progression and relapses. In fact, since many of these food supplements can exert a dual antioxidant/pro-oxidant role, it is necessary to balance these effects to optimize efficacy and limit the toxicity of current antitumoral approaches.

The scientific data collected in this review explain why a fundamental change in the use of supplements is necessary in the treatment of cancer patients. In fact, clinical evidence has shown that the administration of antioxidants in cancer patients may also induce adverse effects in terms of cancer progression.

Based on these considerations, this adjuvant approach represents an important component in the fight against cancer, and greater integration of nutritional, clinical, and molecular studies will be essential to develop more effective preventive and therapeutic strategies.

## Figures and Tables

**Figure 1 antioxidants-14-01261-f001:**
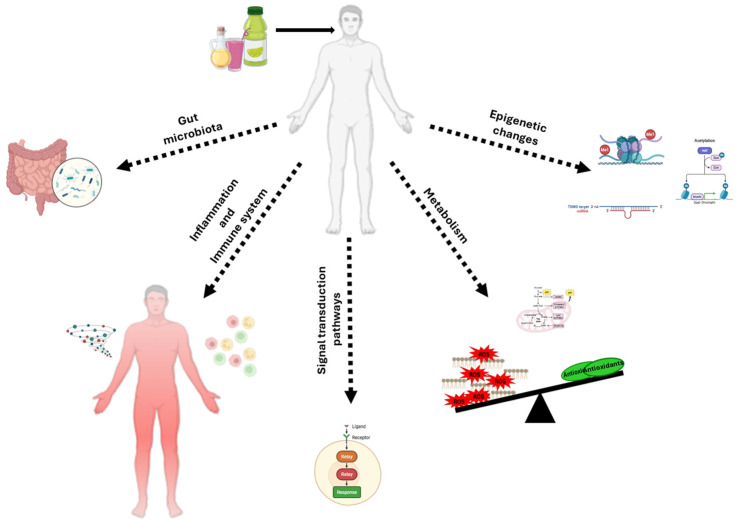
Mechanisms of action of antioxidant supplementation in cancer prevention and therapy. This figure summarizes the most described biological mechanisms through which antioxidant supplementation may influence cancer development, progression, and therapy response. Pleiotropic antioxidant compounds can modulate the following: (i) gut microbiota composition through the production of anticancer metabolites and the reinforcement of intestinal homeostasis, particularly in colorectal cancer; (ii) immune system responses, potentially enhancing anti-tumor immunity or reducing inflammation-induced tumor promotion; (iii) redox-sensitive signaling pathway activity, regulating cell proliferation, apoptosis, autophagy, angiogenesis, and metastasis formation; (iv) cell metabolism, by enhancing oxidative stress and influencing glycolytic metabolism of cancer cells; (v) epigenetic regulation, including modulation of DNA methylation, histone modifications, and particularly miRNA expression, leading to changes in oncogene/oncosuppressor gene activity.

**Table 1 antioxidants-14-01261-t001:** Clinical trials in which antioxidant supplementation was effective in counteracting cancer progression. The efficacy of supplemental antioxidants was evaluated and quantified by comparing clinical outcomes between patients treated with antioxidants and those receiving placebo or standard care, as reported in the original clinical studies.

Cancer Type	Antioxidants	Conventional Therapy	Patients Enrolled (No.)	Ref.
Adenomatous colorectal polyps	- Curcumin (a daily dose of 1 g of capsule 20% curcumin = 200 mg curcumin/daily) - Anthocyanins (a daily dose of 1 g of capsule 36% anthocyanins = 360 mg anthocyanins/daily)	Candidate for surgical removal of precancerous lesions	35 (pre-cancerous lesions)	[14]
Breast cancer	- Curcumin (180.34 ± 9.46 mg per capsule) - Resveratrol (64.51 ± 6.08 mg per capsule) - Lignans (32.44 ± 0.81 mg per capsule) - Isoflavones (17.80 ± 0.99 mg per capsule) - Hydroxycinnamic acid derivatives (1.34 ± 0.18 mg per capsule)	Candidate for mastectomy	39 (Tis-IIb)	[15]
	- Curcumin	Tamoxifen	44	[16]
	- Flavonoids (30.62 ± 20.99 mg/day in subjects with cancer recurrence; 57.11 ± 41.08 mg/day in subjects without cancer recurrence)	Radiation therapy (RT) Chemotherapy (CT)	572 (0-IIb)	[17]
	- Vitamin D (100,000 IU every 3 weeks from day 1 of cycle 1 to day 1 of cycle 5)	Anthracycline and taxane	44 (early breast cancer)	[18]
Cervical carcinoma	- Artemisin derivatives (100 mg/day 1st week; 200 mg/day for other 3 weeks)	-	10 (metastasized cervical carcinoma)	[19]
	- Vitamin A (80,000 IU for 8 h for 64 weeks)	NAC + cisplatin + paclitaxel	30 (advanced cervical carcinoma)	[20]
	- Vitamin D (50000 IU every 2 weeks for 6 months)	-	58 (CIN2/3)	[21]
Colorectal cancer	- Curcumin (2 g oral curcumin/day)	FOLFOX (folinic acid + fluorouracil + oxaliplatin)	27 (metastatic colon cancer)	[22]
	- Vitamin D (40,000 IU/day or 400 IU/day) until disease progression	FOLFOX6 + Bevacizumab	139 (metastatic cancer)	[23]
Cutaneous melanoma	- Vitamin D (100,000 IU monthly)	-	500 (stage IB-III)	[24]
Gastric cancer	- Vitamin D (post-operative supplementation 2000 IU/day)	-	417 (stage I-III)	[25]
Glioblastoma	- Artemisin derivatives (100 mg twice daily)	RT and temozolomide (100 mg/m^2^ day or 50 mg/m^2^ day plus lomustine 40 mg/day)	20 (WHO grade 2–4)	[26]
Head and neck cancer	- Resveratrol (400 mg for 12 weeks alongside home enteral nutrition)	-	72 (stage I–IV)	[27]
Hepatocellular carcinoma	- CPT (curcumin, piperine, and taurine) (5 g curcumin capsules, 10 mg piperine, and 0.5 mg taurine taken daily for three consecutive months)	Candidate for TACE (trans-arterial chemoembolization)	20 (intermediate to advanced stages)	[28]
Infratentorial meningioma, brain tumor	- Vitamin C	Oncothermia, ozone therapy, hydrogen inhalation, time-restricted feeding, hydrotherapy, biologicals, acupuncture, yoga, nutritional supplements, pulsed-electromagnetic field, hydrosun, EDTA chelation, coffee, enema, cold abdominal pack, hot foot bath	1 (case study)	[29]
LHNC (locally advanced head and neck cancer)	- Curcumin (8 g every hourly)	External Beam Radiation Therapy (EBRT)	60 (stage III/IV)	[30]
Metastatic duodenal cancer with an atypical KRAS mutation A59T	- Vitamin C	FOLFOX	1 (case study)	[31]
Myeloma	- Vitamin C (500 mg daily during the last 2 cycles)	5-Azacytidine (100 mg/m^2^ for 5 days in 28 day cycles for a total of 3 cycles)	20 (9 high-risk myelodysplastic syndrome)	[32]
	- Vitamin C (50–80 mg/kg/day, days 0–9)	DCAG (15 mg/m^2^ of decitabine (days 1–5) and 300 μg/day of granulocyte colony-stimulating factor (G-CSF, days 0–9) for priming its combination with 10 mg/m^2^ of cytarabine (q12h, days 3–9), 8 mg/m^2^ of aclarubicin (days 3–6))	7 acute myeloid leukemia, 4 chronic myelomonocytic leukemia	[33]
Ovarian cancer	- Indole-3-carbinol (400 mg/daily) - I3C + EGCG (400 mg/daily + 400 mg/daily respectively)	Neoadjuvant platinum-taxane, surgery, adjuvant platinum-taxane	284 (stage III–IV)	[34]
Prostate cancer	- Selenium (200 mg/day of selenium in 0.5-g high-selenium yeast) vs. placebo	-	927 (no cancer at baseline; 64 prostate cancer T1–T2, diagnosed during the study, with 42 in the control group and 22 in the treatment with selenium supplementation group)	[35]
	- Methionine (200 µg + VitE (400 IU) + Vit C (250 mg) daily for 3 to 6 weeks)	-	48 (T1c/T2)	[36]
Thoracic cancer (breast, lung, esophageal)	- EGCG (from 660 to 2574 μmol/L sprayed on the irradiated area)	RT	19 (severe radio-induced dermatitis in cancer stage II–IV)	[37]

**Table 4 antioxidants-14-01261-t004:** Redox signaling pathways modulated by antioxidant supplementation: in vitro studies.

Compound	Tested Doses	Cancer Type	Combined Treatment	Mechanism/Signaling Pathways	Ref.
Artemisin derivatives	30–60 µg/mL (24 h)	Colon cancer	-	- PDK1, pAkt, MDM2 downregulation	[132]
Curcumin	14–33 μM	Breast cancer	Doxorubicin	- Caspase 3-mediated apoptosis - ROS increase - mtDNA fragmentation - Mitochondrial membrane potential reduction - Doxorubicin sensitization - BCL2 gene expression suppression - BAX, BAK, BIM, PUMA gene overexpression - P53 gene upregulation - Chromatin condensation - DNA damage	[133]
Melatonin	12–50 μM (48 h)	Oral squamous cell carcinoma	-	- PD-1 and PD-L1 downregulation in peripheral blood mononuclear cells lysates	[134]
	1 nM (6 days)	Breast cancer	-	- p38MAPK downregulation - MMP-2/MMP-9 inhibition	[135]
	1 mM (24 h)	Breast cancer	-	- OCT4, N-Cadherin, and vimentin downregulation	[136]
	10 nM–2 mM (24–48 h or up to 14 days)	Prostate cancer	-	- IGF-1 and SIRT1 downregulation	[137]
	1–4 mM (24–48 h)	Glioblastoma	-	- SIRT3/Akt inhibition	[138]
	1–2 mM (72 h)	Hepatocellular carcinoma	-	- EMT inhibition - RAF1, MAPK, Snail, and Bcl-2 downregulation	[139]
	0.01–1 mM (48 h)	Triple-negative breast cancer	-	- IGF-IR and VEGF downregulation	[140]
	1.6–4 mM (48 h)	Ovarian cancer	-	- ERK1/2, Akt, STAT3/5, JNK, CREB, and p38MAPK downregulation	[141]
	0–10 mM (24–72 h)	Ovarian cancer	-	- ZEB1/2, Snail, and vimentin downregulation - E-Cadherin upregulation - MMP9 upregulation	[142]
	0–1 mM (48 h)	Colon cancer	0–150 µM 5-FU (48 h)	- PI3K/Akt downregulation - NF-kB/iNOS Inhibition	[143]
Resveratrol	5–40 µM (24–72 h)	Cervical cancer	-	- p53, Bax, and p16 upregulation	[144]
	5–200 µM (12–72 h)	NSCLC	-	- NGFR and AMPK-mTOR upregulation	[145]
	10–200 µM (24 h)	NSCLC	-	- p-Akt and p-mTOR downregulation	[146]
	1–16 µg/mL (24 h) (resveratrol- loaded nanoform.)	Hepatocellular carcinoma	-	- Bax, caspase-8, -9, and -3 upregulation - PI3K, Akt, and VEGF downregulation	[147]
	10–200 mg/L (1–7 days)	Gastric cancer	0.5–10 mg/L doxorubicin (1–7 days)	- Akt inhibition - Reversion of EMT	[148]
	0–200 µM (24 h)	Bladder cancer	-	- p21 and Bcl2 upregulation - Rb, cyclin D, CDK4, VEGF, and FGF-2 downregulation - p38MAPK and caspase 3 activation - Akt inactivation	[149]
	100µM (24–72 h)	Ovarian cancers	-	- Akt and STAT3 inhibition	[150]
	0–200 µM (24–72 h)	Pancreatic adenocarcinoma	0–20 µM gemcitabine (24–72 h)	- NAF-1 inhibition - Nrf2 activation	[151]
	2, 3.5 (IC50 concentration) and 5 µg/mL Res-Nano (48 h)	Oral squamous cancer	-	- Apoptosis - Cell proliferation, cell migration, and metastasis inhibition - p53-dependent and -independent activation of p21 - Disruption of the β-catenin-GLI-1 complex - Degradation of β-catenin and GLI-1 in cytoplasm - Wnt/β-catenin and Hh/GLI-1 signaling pathway downregulation - TCF/LEF and GLI-1 reporter activity inhibition	[152]
	25 µM (72 h)	Prostate cancer	Neocuproine, desferrioamine mesylate, histidine thiourea, superoxide dismutase, catalase, copper supplementation, small interfering RNA against CTR1	- Pro-oxidant effect in presence of copper ion - Mobilization of endogenous intracellular copper - DNA damage - Apoptosis-like cell death - Reduction of the expression/activity of copper transporter CTR1	[153]
Vitamin A	3.33 µM ATRA (24–72 h)	Breast cancer	-	- DOK1 upregulation - PPARγ activation	[154]
	10–80 µM ATRA (48 h)	Colorectal cancer	-	- MLCK and MCL downregulation - ERK1/MAPK inhibition	[155]
	0.1–10 µM ATRA (24–72 h)	Breast cancer	-	- Wnt/β catenin downregulation	[156]
	10 µM ATRA (72 h)	Hepatocellular carcinoma	-	- N-Cadherin, vimentin, Snail, and Twist downregulation - CK18 and E-Cadherin upregulation	[157]
	10 µM 13-cis retinoic acid (3–9 days)	Neuroblastoma	-	- CRABP2, NEFM, ICAM1, and PLAT upregulation	[158]
	0–10 µM 9-cis retinoic acid (24–96 h)	Cutaneous T-cell lymphoma	-	- RARα upregulation - RARγ and RXRα downregulation - JAK/STAT inhibition	[159]
	1–10 µM (24–48 h)	NSCLC	0.5 µM doxorubicin (24–48 h)	- Downregulation of RARB, CRABP2, and CYP26B1 - Inhibition of EGFR	[160]
Vitamin C	1 mM (72 h)	Acute myeloid leukemia	1 µM deazaneplanocin-A (72 h)	- PHGDH and Bcl2 downregulation	[161]
	0–1 mM (7–24 h)	Breast cancer	-	- AMPK upregulation	[162]
	4–5 mM (2–7 days)	Pancreatic cancer	-	- Wnt/β-catenin downregulation - EMT inhibition	[163]
	0–10 mM (24 h)	Cervical carcinoma	13.8 µM cisplatin or 0.4 µM doxorubicin (24–72 h)	- p53, p21, cyclin D, survivin, LC3BII, cyclin B1 downregulation	[164]
	0–3.5 mM (48 h)	Ovarian cancer	0.06–0.4 mM carboplatin	- ATM/AMPK activation - mTOR inhibition	[165]
	5–10 mM (2–20 h)	Colon cancer	0.4 µM cetuximab (12 h)	- MAPK/EGFR inhibition - PKM2 downregulation	[166]
	0.01 mM–200 mM	Endometrial cancer	Paclitaxel, ipatasertib, N-acetylcysteine	- Pro-oxidant activity - PTEN/AKT/mTOR pathway modulation - G1 phase cell cycle arrest - Apoptosis - DNA damage - Inhibition of adhesion, invasion, and migration - Autophagy - Mitochondrial dysfunction - MAPK pathway modulation	[167]
	1–20 mM	Gastric cancer	L-Buthionine-sulfoximine (BSO), iron chelator (2,2′-bipyridyl)	- Hydrogen peroxide generation - Oxidative stress - DNA synthesis suppression - DNA damage - Caspase-3/7 activation - Apoptosis - Mitochondrial dysfunction - ATP level decrease	[168]
Vitamin D	10 nM 1,25-dihydroxyvitamin D3 (24–72 h)	Cutaneous melanoma	-	- PTEN upregulation - AKT downregulation	[169]
	10 µM calcitriol (24–48 h)	Breast cancer	-	- TIMPs upregulation - MMPs 2 and 9 downregulation - VEGF, TGF-β, and amphiregulin downregulation	[170]
	0.5–1 µM (24 h)	Breast cancer	-	- mTOR suppression - AMPK activation - EMT inhibition	[171]
	7.78–500 µM (24–72 h)	Breast cancer	-	- p53 upregulation - Cyclin-D1 and Bcl2 downregulation	[172]
	0–1000 nM calcitriol (48 h)	Colorectal cancer	-	- Wnt/β-catenin downregulation	[173]
	100 nM calcitriol or EB1089 (synthetic analog) (48 h)	Pancreatic cancer	-	- FOXM1 and Cyclin D, Skp2, c-Myc, CD44, β-catenin, and c-Met downregulation	[174]
	0.1–1 µM calcitriol or MART-10 (synthetic analog) (48 h)	Breast cancer	-	- E-cadherin upregulation - N-cadherin and P-cadherin downregulation - EMT repression	[175]
	0.1–1 µM calcitriol or MART-10 (synthetic analog) (48 h)	Pancreatic cancer; anaplastic thyroid cancer	-	- MMP-2 and MMP-9 downregulation - EMT repression	[176,177]
Vitamin E	0–30 µM β-Tocotrienol (24 h)	Lung cancer; prostate cancer	-	- JACK/STAT 3 inhibition - Reduced phosphorylation of m-TOR, Met, and Akt	[123]
	0–30 µM δ-Tocotrienol (24–48 h)	Lung cancer; prostate cancer	-	- TCF4-STT3a/STT3b axis inhibition	[124]
	5–20 μg/mL δ-Tocotrienol (24–48 h)	Melanoma	-	- Activation of PERK/p-eIF2α/ATF4/CHO, IRE-1α, and caspase-4 ER stress-related branches	[178]
	5–20 μg/mL δ-Tocotrienol (24–48 h)	Prostate cancer	-	- p-JNK and p-p38MAPK upregulation	[179]
	1–5 μg/mL γ-Tocotrienol (7–8 days)	Breast cancer; colon cancer; cervical cancer	-	- SHP1 upregulation - SHP2 and of RAS/ERK downregulation	[180]
	10–80 μM γ-Tocotrienol (6–24 h)	Prostate cancer	-	- RAF/RAS/ERK, caspase 9 and 3 activation - phospho-c-Jun upregulation	[181]
	5–40 μM γ-Tocotrienol (12–60 h)	Prostate cancer	-	- Caspase 3 activation - Akt, MSK, ERK 1/2, and p27 downregulation	[123]
	10–50 μM β-Tocotrienol (24–48 h)	Breast cancer	-	- Caspase 3 and PARP activation - Bax/Bcl2 ratio increase - p-PI3K and p-GSK-3 downregulation	[182]
	2–12 μM γ-Tocotrienol (96 h)	Breast cancer	-	- p-Akt downregulation - AMPK activation - FoxO3 inactivation	[183]
	5–50 μM γ-Tocotrienol (24–120 h)	Colorectal cancer	20 μM capecitabine (18–24 h)	- cIAP-1, cIAP-2, survivin, c-myc, and cyclin D downregulation - MMP9, VEGF, ICAM-1, and CXCR4 downregulation	[184]
	0–100 μM γ-Tocotrienol (24–72 h)	Pancreatic cancer	-	- EMT inhibition - MMP9 and VEGF downregulation	[185]
	50 μM γ-Tocotrienol (24 h)	Liver cancer	-	- Akt/mTOR inhibition	[186]
	0–50 μM γ-Tocotrienol (24–72 h)	Gastric cancer	10 μM capecitabine	- Inhibition of constitutive and capecitabine-induced NF-kB	[187]
	0–40 μM γ-Tocotrienol (24–72 h)	Breast cancer	-	- Activating transcription factor 3 upregulation	[188]
	50–150 μM γ-Tocotrienol (24–48 h)	Bladder cancer	0.08 μM gemcitabine (48 h)	- p21 and p27 upregulation - Cyclin D1 downregulation - STAT3 signaling suppression	[189]
	0–100 μM γ-Tocotrienol (24 h)	Breast cancer	-	- p21 and p27 upregulation - CyclinD1 downregulation - STAT3 signaling suppression	[190]
	0–30 μM γ-Tocotrienol (20 and 72 h)	NSCLC	-	- MMP-9 downregulation - uPA/Notch-I and NF-kB activity inhibition	[191]

**Table 5 antioxidants-14-01261-t005:** Redox signaling pathways modulated by antioxidant supplementation: in vivo studies.

Compound	Tested Doses	Cancer Type	Combined Treatment	Mechanism/Signaling Pathways	Ref.
Artemisin derivatives	40 mg/kg/day (21 days)	Colon cancer	-	- Decrease in PDK1, pAkt, and pMDM2	[132]
NAC	1 g/liter (1 week)	Lung cancer	-	- p53 downregulation	[4]
	100 mg/kg (thrice weekly for 5 weeks)	Pancreatic cancer	100 mg/kg gemcitabine (thrice weekly for 5 weeks)	- Block of NF-kB activation induced by gemcitabine	[192]
	100 mg/kg	Glioblastoma	-	- Notch2 degradation	[193]
Melatonin	40 mg/kg/day (5 days a week; for 3 weeks)	Breast cancer	-	- VEGF receptor 2 downregulation	[194]
	1 mg/kg/day (41 days)	Prostate cancer	-	- Nrf2, HIF-1α, and pAkt upregulation	[195]
	10 and 20 mg/L (18 weeks)	Prostate cancer	-	- IGF-1 and SIRT1 downregulation	[137]
	25 mg/kg/day (32 days)	Thyroid cancer	Irradiation 2Gy (2 times weekly for 25 days)	- PTEN upregulation	[196]
	5.50 mg/kg/day (4 weeks)	Glioblastoma	-	- SIRT3/Akt pathway inhibition	[138]
	40 mg/kg (5 days/week for 3 weeks)	Hepatocellular carcinoma	-	- RAF1, MAPK, Snail, and Bcl-2 downregulation	[139]
	40 mg/kg (21 days)	Triple-negative breast cancer	-	- HIF-1α and VEGF downregulation	[140]
	200 mg/kg (3 days/week for 4 weeks)	Oral squamous cell carcinoma		- ABCB1 and ABCB4 downregulation	[197]
	2.5 µg/day	Breast cancer	Doxorubicin (6 mg/kg/day)	- pERK1/2 suppression - Akt, NF-kB, p-PDK1, PKCα, PKCδ, p-STAT3 downregulation	[198]
	10 mg/kg/day (27 days)	Breast metastatic cancer	Doxorubicin (1.25 or 2.5 or 5 mg/kg/day; 21 days)	- p65 and p-STAT3 downregulation	[199]
	25 mg/kg/day (17 days)	Colon cancer	5-FU (20 mg/kg/day) (17 days)	- p-p65, p-Akt and iNOS downregulation - E-Cadherin upregulation	[143]
Resveratrol	15 mg/kg (3 days/week for 5 weeks)	Cervical cancer	-	- Inhibition of HPV E6 and HPV E7 expression - p53, Bax, and p16 upregulation	[144]
	2.5–10 mg/kg (resveratrol-loaded nanoform.) (1 inject. every days for 12 days)	Breast cancer	-	- p53 and caspase-3 upregulation - Bax/Bcl2 ratio downregulation	[200]
	80 mg/d (14 days)	Colon cancer	-	- myc and cyclin D1 increased expression	[100]
	5 or 50 mg *trans*-resveratrol (twice daily for 12 weeks)	Breast cancer	-	- Suppressing methylation of the RASSF-1α gene and lower -promoting PGE2 levels	[84]
	50 mg subcutaneous pellet (once a month for 8 months)	Breast cancer	-	- NRF2 upregulation - Prevention of E2-mediated inhibition of detoxification genes AOX1 and FMO1 - Apoptosis induction	[201]
	50 mg/kg (once a week for 4 weeks)	Gastric cancer	3 mg/Kg doxorubicin	- PTEN activation - EMT reversion	[148]
	20 mg/kg/day (4 weeks)	Bladder cancer	-	- VEGF and FGF-2 downregulation	[149]
	50 mg/kg daily (6 weeks)	Colon cancer	Resistance training	- mTORC and AMPK inactivation - LC3BII/I ratio downregulation	[202]
Curcumin	2 capsules 1 gm (every 8 h daily)	LNHC	ERBT	- Suppression of the activity of NF-kB - Prevention of LOX and COX-2 activation - TGF-β1 upregulation	[30]
	From 8 to 162 mg·kg^−1^·day^−1^, for 0.05% and 1% diets	Colorectal cancer	-	- Reduction of aberrant localization of β-Catenin	[109]
Curcumin and anthocyanin	160 mg with 2 daily administrations and ranging up to a daily dose of 10.8 g for 7 days of treatment	Adenomatous polyps	-	- Increase in IL-6, IL-10, and IGFBP-3	[14]
Isoflavones, lignans, phytoestrogens	100 mg isoflavones and 100 mg of lignans and thus 200 mg of phytoestrogens	Prostate cancer	Candidate to prostatectomy	ERβ	[87]
Indole 3 carbinole	200 mg for 28 days and then 400 mg for 28 days	Breast cancer	-	- Increase in E2 2-OH/16⍺-OH (2-hydroxy/16⍺- hydroxy estrogen) CYP1A2	[203]
	IC 200/400 mg per day for 28 days	Cervical dysplasia	-	- Increase in E2 2-OH/16⍺-OH (2-hydroxy/16⍺- hydroxy estrogen) and CIN (cervical intraepithelial neoplasia)	[203]
Vitamin A	1 µM (twice/week for 3 weeks)	NSCLC	-	- EGFR downregulation - Wnt/CTNNB1 inhibition	[160]
Vitamin C	4 g/kg/day + 60 mg/kg/day aspirin (90 days)	Liver cancer	0.72 mg/rat/day doxorubicin (once a week for 90 days)	- p53 upregulation - ALT, AST, albumin, TBIL, AFP, and CA19.9, IL-6, CASP3, CASP8, and BAX downregulation	[204]
	4 g/kg (twice daily for 26 days)	Liver cancer	-	- Stemness gene upregulation	[205]
	4 g/kg (twice daily for 25 days)	Ovarian cancer	20 mg/kg carboplatin or 5 mg/kg paclitaxel (once per week)	- ATM/AMPK activation - mTOR inhibition	[165]
	4 g/kg/day (45 days)	Pancreatic cancer	40 mg/kg gemcitabine (every 3 days for 45 days)	- Collagen and CK-19 upregulation - Snail downregulation	[206]
	Oral administration: 1.11 g/kg/day (4 weeks) Intraperitoneal (IP) injection: 4 g/kg, twice a day for 4 weeks	Endometrial cancer	Paclitaxel, ipatasertib, N-acetylcysteine	- PTEN/AKT/mTOR pathway inhibition - G1 phase cell-cycle arrest - Apoptosis - DNA damage - Inhibition of adhesion, invasion, and migration - Autophagy induction - Mitochondrial dysfunction	[167]
Vitamin D	500 IU/rat/day (3 days/week)	Colorectal cancer	5-fluorouracil (12 mg/kg/day for 4 days, then 6 mg/kg for 4 days)	- Wnt/β-catenin, iNOS, COX-2, and HSP-90 downregulation - DKK-1, TGF-β1, TGF-βR2, and smad4 upregulation	[207]
	1–10 µg/kg/day PRI-2191 or PRI-2205 (Vit D analog) (3 times a week for 4 weeks)	Lung cancer	50 mg/kg/day imatinib (13 days)	- VEGF and Bcl2 downregulation - p53 upregulation	[208]
	5300 IU/Kg Vit D3 or 25 ng calcitriol (3 times a week for 6 weeks)	Breast cancer	-	- Repression of estrogen receptor, aromatase and Cox-2 expression - PGE2 downregulation - Adiponectin receptor upregulation	[209]
Vitamin E	200 mg/kg δ-tocotrienol (twice daily for 12 months)	Pancreatic cancer	-	- MEK/ERK, PIK/Akt, NF-kB/p65, and Bcl2 downregulation - Bax induction - Caspase 3 activation	[210]
	100 mg/kg γ-Tocotrienol (5 times/week for 2 weeks)	Colorectal cancer	60 mg/kg capecitabine (twice week for 2 weeks)	- c-myc, cyclin D1, MMP9, VEGF, ICAM-1, and CXCR4 downregulation	[184]
	200 mg/kg/day γ-Tocotrienol (4 weeks)	Pancreatic cancer	100 mg/kg gemcitabine (twice a week for 4 weeks)	- MMP9 downregulation	[185]
	3.25 mg γ-tocotrienol (5 days/week)	Liver cancer	-	- VEGF and CD31 downregulation - Caspase 3 activation - Akt and tumor-induced angiogenesis inhibition	[186]
	1 mg/kg γ-tocotrienol (3 times/week for 4 weeks)	Gastric cancer	60 mg/kg capecitabine (twice week for 4 weeks)	- CD31 downregulation - Inhibition of constitutive NF-kB	[187]
Sulforaphane (Broccoli natural sprout)	26% (*w*/*w*) + Inulin 2% (*w*/*v*)	Breast cancer	-	- AKT/PI3K/mTOR suppression - Cell-cycle arrest - Caspase 3 and caspase 7 activation - CDK2 and CDK4 inhibition	[112]

## Data Availability

No new data were created or analyzed in this study. Data sharing is not applicable to this article.

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
