# Peer review of "Antioxidant Food Supplementation in Cancer: Lessons from Clinical Trials and Insights from Preclinical Studies"

_antioxidants, 2025, doi:10.3390/antiox14101261_

Round 1

Reviewer 1 Report

Food antioxidant supplementation has been widely proposed for cancer prevention and adjuvant therapy, taking into consideration the pleiotropic role of antioxidants. Herein, particular attention is given to recent clinical trials based on the use of dietary supplements in cancer patients, both as monotherapy and in combination with standard treatments, exploring both their potential benefits and risks. This review focuses on the efficacy of the most important food antioxidants, highlighting how their actions can change from beneficial to neutral to adverse depending on factors such as cancer type, dose, timing of administration, and the patient's antioxidant status. As a result, the findings of clinical trials using antioxidants as stand-alone or adjuvant cancer therapies are often contradictory, with clinical benefits being more evident in patients with a baseline antioxidant deficiency and in inhibiting or ameliorating the often-debilitating side effects of anticancer therapies.

The updated manuscript has taken on board most, if not all, of the issues raised in the initial reviews and now presents a more coherent and instructive treatise on the potential benefits and detriments of antioxidants alone or as adjuvants in cancer therapies or in mitigating the toxic side effects of therapies. The review also now covers more on the host, dietary, microbial, or environmental factors that may be critical in defining whether antioxidants will or will not have beneficial effects. The review now provides a good overview for clinicians and a guide for effective design of future studies in the use of specific antioxidants as supplements in cancer therapy.

The authors have dealt with all issues in a thorough and robust manner.

Author Response

Reviewer #1

Food antioxidant supplementation has been widely proposed for cancer prevention and adjuvant therapy, taking into consideration the pleiotropic role of antioxidants. Herein, particular attention is given to recent clinical trials based on the use of dietary supplements in cancer patients, both as monotherapy and in combination with standard treatments, exploring both their potential benefits and risks. This review focuses on the efficacy of the most important food antioxidants, highlighting how their actions can change from beneficial to neutral to adverse depending on factors such as cancer type, dose, timing of administration, and the patient's antioxidant status. As a result, the findings of clinical trials using antioxidants as stand-alone or adjuvant cancer therapies are often contradictory, with clinical benefits being more evident in patients with a baseline antioxidant deficiency and in inhibiting or ameliorating the often-debilitating side effects of anticancer therapies.

The updated manuscript has taken on board most, if not all, of the issues raised in the initial reviews and now presents a more coherent and instructive treatise on the potential benefits and detriments of antioxidants alone or as adjuvants in cancer therapies or in mitigating the toxic side effects of therapies. The review also now covers more on the host, dietary, microbial, or environmental factors that may be critical in defining whether antioxidants will or will not have beneficial effects. The review now provides a good overview for clinicians and a guide for effective design of future studies in the use of specific antioxidants as supplements in cancer therapy.

Detailed comments

The authors have dealt with all issues in a thorough and robust manner

We are pleased to know that the Reviewer appreciated the updated manuscript.

Reviewer 2 Report

This review provides understanding to therapeutic effects of various antioxidants therapies in cancer treatment.

My comments are as follows:

  1. To provide methodology of literature search covering the information within 25 years.
  2. Table 1 to Table 3 : Arrangement can be based on alphabetical order of cancer type.
  3. Candidate to surgery - this can be replaced by types of surgery eg mastectomy
  4. Caption for Table 2 - repetitive words
  5. To check the use of capital and small letter - Breast Cancer --- Breast cancer
  6. Table 1, 2 and 3 should be discussed separately in the text with relevant citations
  7. The information starting from 3.1 is confusing and can be rearranged. Based on my understanding the clinical and preclinical studies are discussed in section 3.1 onwards. e.g.

3.1 Effects on gut microbiota

Clinical studies (numbering is not needed)

Text

Preclinical studies (numbering is not needed)

Text

8. Include a section of Future Perspectives before the Conclusion

-

Author Response

This review provides understanding to therapeutic effects of various antioxidants therapies in cancer treatment. My comments are as follows:

  • To provide methodology of literature search covering the information within 25 years.

Literature contents were selected from PubMed, taking into consideration those published in the last 25 years. The keywords used were: antioxidant supplementation and i) clinical trials ii) in vitro studies; iii) in vivo studies. Clinical trials examined were those with a number of participants equal to or greater than 10. Furthermore, for completeness of information, two case studies were also included.

2) Table 1 to Table 3 : Arrangement can be based on alphabetical order of cancer type.

We thank the reviewer for this valuable suggestion, which makes the tables easier to understand. In accordance with this request, Tables 1, 2 and 3 have been arranged on alphabetical order of cancer type. Moreover, the order of the papers cited has also been updated accordingly.

3) Candidate to surgery - this can be replaced by types of surgery eg mastectomy

As required, in the tables, candidate to surgery has been replaced by adding the types of surgery.

4) Caption for Table 2 - repetitive words

We apologize for the mistake, the repetitive words have been deleted.

5) To check the use of capital and small letter - Breast Cancer --- Breast cancer

As required, the use of capital and small letters has been checked and the corrections have been made.

6) Table 1, 2 and 3 should be discussed separately in the text with relevant citations

We sincerely thank the reviewer for the suggestion to discuss in detail the data presented in Tables 1-3, including individual references. We fully acknowledge that this is a legitimate and valuable approach. However, we have chosen not to provide a separate discussion for each table to avoid redundancy and excessive repetition of concepts already summarized in the text. In fact, as you can see, in order to make easier to find the information, the most salient results reported in Tables 1-3 have been discussed and reported with the relevant references in the paragraphs 2.1 - 2.10 in which they have been divided based on the antioxidant compound examined.

7) The information starting from 3.1 is confusing and can be rearranged. Based on my understanding the clinical and preclinical studies are discussed in section 3.1 onwards. e.g.

We fully appreciated this suggestion and as required, the information has been rearranged.

8) Include a section of Future Perspectives before the Conclusion

We agree with the Reviewer and, as suggested, a Future Perspectives section has been added before the Conclusion.